# Screening and Regulation Mechanism of Key Transcription Factors of *Penicillium expansum* Infecting Postharvest Pears by ATAC-Seq Analysis

**DOI:** 10.3390/foods11233855

**Published:** 2022-11-29

**Authors:** Lina Zhao, Yuling Shu, Sihao Quan, Solairaj Dhanasekaran, Xiaoyun Zhang, Hongyin Zhang

**Affiliations:** School of Food and Biological Engineering, Jiangsu University, Zhenjiang 212013, China

**Keywords:** pear fruit, *Penicillium expansum*, transcription factors, MAPK signaling pathway, infection

## Abstract

Transcription factors play a key role in *Penicillium expansum* infection process. Although the crucial characteristics of some transcription factors of pathogenic fungi have been found, many transcription factors involved in *P. expansum* infections have not been explored and studied. This study aimed to screen the transcription factors of *P. expansum* involved in postharvest pear infections by ATAC-seq analysis and to analyze the differentially expressed peak-related genes by GO enrichment and KEGG pathway analysis. Our results found the up-regulation of differentially expressed peak-related genes involved in the MAPK signaling pathway, pentose phosphate pathway, starch and sucrose metabolism, and pentose and glucuronate interconversions. Our study especially confirmed the differential regulation of transcription factors MCM1, Ste12 and gene *WSC* in the MAPK signaling pathway and *PG1*, *RPE1* in the pentose and glucuronate interconversions pathway. These transcription factors and related genes might play an essential role in pear fruit infection by *P. expansum*. RT-qPCR validation of twelve expressed peak-related genes in *P. expansum* showed that the expression levels of these twelve genes were compatible with the ATAC-Seq. Our findings might shed some light on the regulatory molecular networks consisting of transcription factors that engaged in *P. expansum* invasion and infection of pear fruits.

## 1. Introduction

Pears are one of the most important traditional fruits, rich in nutrients and high in demand worldwide. However, pears are highly vulnerable to various pathogenic fungi during harvest, packaging, storage, and transportation, which results in significant financial losses. According to statistics, fungal infection due to physiological diseases and mechanical damage cause about 30% economic loss in the pear fruit industry [1]. The most prevalent postharvest diseases of pear fruit are blue mold decay caused by *Penicillium expansum* and gray mold decay caused by *Botrytis cinerea*. Additionally, *P. expansum* produces patulin, a hazardous mycotoxin, which causes neurotoxic, immunotoxic, immunosuppressive, genotoxic, teratogenic and carcinogenic effects in animals and humans [2]. Our previous research investigated the physiological processes of *P. expansum* pathogenicity, such as pathogen activation, germination, and growth in pear tissues using SEM at basic contamination time intervals [3]. The results suggested that pear fruit infection by *P. expansum* includes germ tube formation through spore germination and secretion of infectious substances; *P. expansum* infection damages the entire pear tissue after 24 h [3]. Our study established the critical roles of pathogenicity factors of *P. expansum* in the decay process. Therefore, it is incredibly important to further study the pathogenic mechanism of pathogenicity factors in *P. expansum* infecting pear fruit.

Pathogenicity factors can be defined as pathogen components that are not necessary for in vitro growth but contribute to disease. In general, these factors can be as diverse as plant cell wall degrading enzymes (CWDEs), effectors (pathogenic secretory proteins), mycotoxins and others [4]. During *P. expansum* infection in apple fruit, the activities of CWDEs, including polygalacturonase and pectin methylesterase, were significantly increased [5]. Levin et al. (2019) analyzed the secretome of *P. expansum* to predict the pathogenicity-related factors and found that the deletion of the PePRT-coding gene resulted in reduced virulence of *P. expansum* in apples [6]. *P. expansum* produces two important mycotoxins, patulin and citrinin. Previous studies have evidenced that patulin represents a colonization factor for *P. expansum* infection in apples, and citrinin is an establishment factor of *P. expansum* colonization in apples [7]. Transcription factors play an essential role in fungal virulence and the infection process. Transcription factor Ste12 of *P. expansum* played a vital role in fungi’s virulence and asexual reproduction during apple fruit infection [8]. Similarly, a novel MADS-box transcription factor PstMCM1-1 of *Puccinia striiformis* f. sp. Tritici was responsible for the fungi’s complete virulence [9].

In recent years, the study of transcription factors has become popular, and some exciting developments have been made. Although many transcription factors of pathogenic fungi have been identified, the information on the transcription factors related to *P. expansum* infecting host fruit is insufficient and needs further excavation and study. In addition, the knowledge behind the molecular mechanisms of *P. expansum* infecting pear fruit is still limited; the research on the pathogenicity factors and their essential roles in pathogenic mechanisms during pear fruit infection is also scarce. Above that, many pathogenic and transcription factors have not been identified, and their important functions are not studied. Therefore, this study aimed to screen the transcription factors of *P. expansum* infecting postharvest pear fruit by assay for transposase-accessible chromatin with sequencing (ATAC-Seq) analysis and analyze the differentially expressed peak-related genes by Gene Ontology (GO) enrichment and Kyoto Encyclopedia of Genes and Genomes (KEGG) pathway analysis. Finally, Real-Time Quantitative PCR (RT-qPCR) verified the accuracy of the ATAC-seq data. This study provides a unique hypothetical establishment on the *P. expansum* invasion and infection component in pear fruit.

## 2. Materials and Methods

### 2.1. Fruits

Pears (*Pyrus pyrifolia* Nakai “Shuijing”) were harvested from an orchard in Zhenjiang, Jiangsu province. The pears were selected randomly based on commercial maturity, uniform size and without mechanical damage. The pears were washed with tap water, drenched in 0.1% NaClO for 2 min, washed with tap water to remove the remaining NaClO, then dried at room temperature.

### 2.2. Pathogen

*P. expansum* was isolated from the surfaces of rotten pears and stored at −20 °C. The fungus was activated in potato dextrose agar (PDA) plates for 7 days at 25 °C before use [2]. The spores of *P. expansum* were collected and suspended in sterilized water and then adjusted to 1 × 10^7^ spores/mL using a hemocytometer.

### 2.3. Sample Preparation

Three uniform wounds were made at the equator of pears using a sterile puncher, then 30 μL of *P. expansum* spore suspension (1 × 10^7^ spores/mL) was added. The inoculated fruits were placed in a plastic basket and kept in an incubator at a constant temperature and humidity (25 °C, RH 95%) for 24 h. Tissue samples from *P. expansum* inoculated pears were collected at 24 h and named T-1 and T-2. Spores from the *P. expansum* strain cultured in PDA plates for 7 days at 25 °C were collected, suspended in sterilized water, used as the control, and named CK1-1 and CK1-2. The sampling time was fixed, and the sample preparation was done according to Xu et al. [3].

### 2.4. ATAC-seq and Analysis

A transposase-containing Transposition Mix was used to incubate the nuclei suspensions. The DNA was preferentially broken up in open areas of the chromatin when the Transposase entered the nuclei. Adapter sequences were added to the DNA fragments at the end of each strand. The mixture was incubated for transposition reaction at 37 °C for 30 min. Immediately after the transposition, the products were purified using a QIAGEN minielute kit, amplified as described by Buenrostro [10]. The ATAC-seq was performed using Illumina HiSeqTM 4000 (Illumina, San Diego, CA, USA) at Gene Denovo Biotechnology Co. (Guangzhou, China).

#### 2.4.1. Clean Reads Filtering

Reads obtained from the sequencing machines included raw reads containing adapters or low-quality bases. High-quality clean reads were obtained by eliminating the adapters, more than 10% of unknown nucleotides, and low-quality bases with Q value ≤ 20. Reads alignment: The clean reads obtained from each sample were aligned with the reference genome using the Bowtie2 tool (version 2.2.8, Baltimore, MD, USA) [11], and duplicates in all data files were eliminated by Picard. Peak scanning: There was an offset of +4 bps for all reads aligning with the + strand and –5 bps for all reads aligning with the − strand. MACS (version 2.1.2, Boston, MA, USA) was used for the peak calling of shifted, concordantly aligned paired mates with parameters “--nomodel --shift -100 --extsize 200 -B -q 0.05” [12]. MACS is a computational method designed to identify read-enriched regions from sequencing data. Based on the unique mapped reads, the p-value of a particular region was calculated by Dynamic Poisson Distribution. Q-values were calculated from the p-values. When the q-value was less than 0.05, the region was termed a peak. Peak-related genes annotation: Peak-related genes were verified based on the genomic location information and gene annotation information by the ChIPseeker tool [13]. Besides, the distribution of peaks on different genomic regions was also evaluated.

#### 2.4.2. Irreproducible Discovery Rate (IDR)

Irreproducible discovery rate (IDR) was used to verify the consistency between biological replicates within an experiment. The reproducible peaks with an IDR value of 0.05 or less were kept [14]. Motif analysis: The DNA binding site for specific transcription factors or histone modifications was not random, but they showed a conserved DNA sequence pattern. The motifs were detected utilizing the MEME suit. MEME-ChIP was used to scan the motifs reliably through peak regions, and then AME was used to validate the existence of any certain motifs. Differential analysis of multi-samples: The DiffBind was used to analyze peak differences across groups [15]. We identified significantly differential peaks with FDR < 0.05 in two comparison groups. Similarly, genes associated with different peaks were annotated, and the GO functions of enrichment analysis and KEGG pathways were identified.

### 2.5. Verification of the Relative Expression Levels of Differentially Expressed Peak-Related Genes in P. expansum by RT-qPCR

Twelve differentially expressed peak-related genes in *P. expansum* were selected to verify the relative expression levels. The primers were designed by Primer 5 and synthesized by Sangon Biotech (Shanghai, China), and the primers used in this study are listed in Table 1. Total RNA was extracted according to the instructions of the Trizol reagent kit (Invitrogen, Carlsbad, CA, USA). HiFiScript gDNA Removal RT Master Mix (CoWin Biosciences, Jiangsu, Beijing, China) was used to reverse the transcription of RNA into cDNA. Then RT-qPCR was carried out using TB Green^®^ Fast qPCR Mix (Taraka Bio Inc., Shiga, Japan) and determined in ABI PRISM 7500 Real-Time PCR System (Applied Biosystems, Waltham, MA, USA) according to the manufacturer’s instructions. The 2^−ΔΔCT^ method was used to calculate the relative expression of genes, and the DOA4 expression was used as an internal control [16]. The experiment was repeated twice, and each treatment was triplicated.

## 3. Results

### 3.1. Chromatin Accessibility Landscape

The chromatin accessibility landscape of CK1 and T samples is shown in Figure 1. The transcription initiation site (TSS) is a highly open region of the genome. The more open it is, the more transcription factors bind to start gene transcription. As shown in Figure 1A, the inserted fragments in the chromatin open region tend to be enriched in the TSS region, verifying the reliability of cutting and data. The high incidence region of chromatin opening was 2KB in front of the promoter region, and it can be seen from Figure 1B that the distribution of peak in each functional region of the gene in the two samples showed a similar distribution pattern, in which the region 2KB upstream of the gene accounts for the majority. The Pearson correlation coefficient of two ATAC-seq replicates of two group samples is shown in Figure 1C; the R-value between the two samples arbitrarily selected in the control group was greater than 0.99, the R-value between the two samples arbitrarily chosen in the experimental group was greater than 0.95, while the R-value between the two samples arbitrarily selected in the control group and the experimental group was less than 0.9, indicating that the parallelism of the treatment was good. The difference between the treatments was significant.

### 3.2. GO Enrichment Analysis of Differentially Expressed Peak-Related Genes in P. expansum

When |log2(Fold Change)| ≥ 1 and FDR < 0.05 were used as the screening criteria, a total of 1017 differentially expressed peak-related genes (DEGs) were identified, with 987 up-regulated DEGs and 30 down-regulated DEGs. A add up to of 541 DEGs were distinguished (|log2 (Fold Change)| ≥ 1.2 and FDR < 0.05), including 528 up-regulated and 13 down-regulated genes, and these DEGs were then analyzed using GO enrichment and KEGG pathway analysis.

As shown in Figure 2A, the DEGs of *P. expansum* were divided into biological process, cell component and molecular function. The biological process included 18 secondary subclasses; among them, biological regulation, response to stimulus and signaling were related to the growth, development and response to stimulation of *P. expansum*. The membrane and membrane part of cell components were associated with the transportation of related substances during the infection of *P. expansum* on pear fruit. The catalytic activity and transporter activity in molecular function were related to the infection ability of *P. expansum*. The third-level GO enrichment analysis was further carried out for the above seven secondary subclasses related to the *P. expansum* infection regulation (Figure 2B). The response to stimulation contained the most third-level classification (7), in which the cellular response to stimulus enriched with the most genes (7), and the enriched DEGs included pheromone receptor transcription factors (MCM1). The subclass of catalytic activity included six tertiary classifications, of which the hydrolase activity enriched the most genes (19), and the enriched DEGs included chitin synthase gene (*CHS8*), glucanase gene (*eglB*) and so on. The transport activity and biological regulation subclasses included three tertiary classifications; the enriched DEGs included the hexose transporter gene (*ght5*) and gluconic acid transport inducer gene (*sge1*). The cell membrane and membrane part subclasses included four and two tertiary classifications, respectively, the intrinsic component of the membrane enriched the most genes (18), and the enriched DEGs included 3-oxidation-5-steroid 4-dehydrogenase gene (*gpsn2*). There was a tertiary classification under the signal subclass, and the enriched DEGs included a heterotrimeric protein subunit gene (*fadA*).

### 3.3. KEGG Enrichment Analysis of Differentially Expressed Peak-Related Genes in P. expansum

As shown in Figure 3A, the differentially expressed peak-related genes were annotated in the KEGG database and enriched in five primary classifications, including environmental information processing, cellular processes, metabolism, genetic information processing and organizational systems. And then, five secondary subclasses of signal transduction, transport and catabolism, carbohydrate metabolism, amino acid metabolism and replication and repair were further analyzed (Figure 3B). The related pathways, including MAPK signaling pathway in signal transduction, pentose phosphate pathway, starch and sucrose metabolism, pentose and glucuronate interconversions and amino sugar nucleotide sugar metabolism in carbohydrate metabolism, arginine biosynthesis in amino acid metabolism, were analyzed further (Figure 4).

As shown in Figure 4A, *P. expansum* infected the pear fruit after 24 h, the MAPK signaling pathway was activated, and the expression levels of multiple genes in *P. expansum* were up-regulated. In the signal transduction pathway induced by pheromone, the expression of gene *GB-1* encoding guanylate binding protein was up-regulated. The gene *MCM1* encoding a pheromone receptor transcription factor and its binding transcription factor Ste12 were significantly up-regulated. In the signal transduction pathway induced by cell wall stress, the expression levels of genes *WSC1,2,3* encoding cell wall integrity and stress response components and *Its3* encoding phosphatidylinositol-4-phosphate-5-kinase were up-regulated. Activated phosphatidylinositol diphosphate promoted the up-regulation of *Mkk1-2* encoding mitogen-activated protein kinase, and the expression of *fksA* encoding 1,3-β-glucan synthase was also up-regulated, which promoted the cell wall remodeling. In the signal transduction pathway induced by high osmolality, the expression of pheromone receptor transcription factor MCM1 bound to zinc finger protein *Msn2,4* was up-regulated and then promoted osmolyte synthesis. In the signal transduction pathway induced by starvation, the expression of transcription factor Ste12 was inhibited by up-regulated expression invasive growth down-regulation factor *Dig1,2*, which could activate flocculating protein Flo11 and form filaments. In pentose and glucuronate interconversions and the pentose phosphate pathway, *PG1*, *RPE1* and pectinesterase genes were up-regulated (Figure 4B).

As shown in Figure 4C, the pathways of starch and sucrose metabolism and amino sugar and nucleotide sugar metabolism were activated in *P. expansum* infecting pear fruit. In the starch and sucrose metabolism pathway, the genes *amy*, *agdA*, *eglB* and *fksA* were up-regulated. In this pathway, starch and other glycogen are synthesized from cyclodextrin by sucrose synthase, which further produces maltodextrin, and synthesize maltose by *amy* (encoding amylase), then synthesizes glucose by *agdA* (encoding maltase). The expression of *eglB* encoding glucanase was up-regulated, which may accelerate the conversion of cellulose to cellobiose and further to glucose. Starch and other glycogen can also produce 1-phosphate-glucose by 1,3-β-glucan synthase gene *fksA* to synthesize glucose. UDP-glucose could be converted to sucrose and then converted to glucose by *agdA*. In the amino sugar and nucleotide sugar metabolism pathway, the expressions of *Chs* and *cbr1* were up-regulated. The up-regulation of *Chs* encoding chitin synthase may promote the synthesis of chitin. UDP-glucose could be converted to N-acetylneuraminic acid and then synthesized cytochrome protein by *cbr1* encoding cytochrome b5, then synthesis of adenosine-5 ‘- monophosphate-N-hydroxyacetylneuraminic acid. In the arginine biosynthesis pathway, the expression of *gdh1* and *DUR1,2* were up-regulated (Figure 4D).

### 3.4. TF Motif Analysis of Differentially Expressed Peak-Related Genes

The differentially expressed peak was enriched with nine transcription factors, as shown in Table 2. They were HAP3 (Heteromeric CCAAT-binding factors), SOK2 (APSES-type DNA-binding domain), STP3 (C2H2 zinc finger factors), SIP4, YPR196W, ECM22, YLR278C, SUT2 and YNR063W (C6 zinc cluster factors), respectively. Five of them control the gene *WSC*, which is responsible for cell wall integrity and components of the stress response; four of the transcription factors control the gene *RPE1*, which encodes ribulose phosphate 3-isomerase; and transcription factor Ste12 was controlled by three of them. According to the related genes in the KEGG pathways, the Gene ID, symbol and description of the transcription factors regulating genes are listed in Table 2.

### 3.5. The Expression Level of Differentially Expressed Peak-Related Genes

The relative expression of the differentially expressed peak-related genes in *P. expansum* was confirmed by RT-qPCR of 12 genes that included 8 up-regulated and 4 down-regulated genes. (Figure 5). The results showed that the expression levels of differentially regulated genes identified in ATAC-seq and RT-qPCR were highly correlated, which confirmed the stability and dependability of the ATAC-Seq results.

## 4. Discussion

Diversified pathogenicity or virulence factors of pathogenic fungi, such as plant cell wall degrading enzymes, effectors (pathogenic secretory proteins), mycotoxin, etc., have been reported so far. Moreover, understanding the molecular regulation of fungal virulence is critical to understanding the nature of plant diseases. Transcription factors (TFs) are sequence-specific DNA-binding proteins that regulate gene expression. TFs can control chromatin and transcription by recognizing specific DNA sequences and directly regulate the expression of downstream functional genes or indirectly affect the expression of downstream genes by regulating the expression of other transcription factors. Although the significant roles of transcription factors in pathogenic fungi have been reported, studies on the transcription factors related to *P. expansum* infecting the hosts are limited and need further investigation. In this study, the *P. expansum* infecting pear fruit was analyzed by ATAC-seq at key time points. The candidate transcription factors related to *P. expansum* infection and their regulated corresponding downstream genes were screened.

MAPK signaling pathway regulates fungal development, growth, pathogenicity and mediating environmental responses [17]. Ste12 and MCM1 are transcription factors in the MAPK signaling pathway, regulated by transcription factor HAP3. HAP3 is an essential regulator of iron homeostasis in fungi, which is necessary for fungal cell survival and important to maintain virulence in various pathogenic fungi [18,19]. Ste12 was first found in *Saccharomyces* cerevisiae and is essential for expressing genes required for mating (those involved in pheromone response) and for genes unrelated to mating [20]. However, the functions of Ste12 vary in different fungi. It has been reported that transcription factor Ste12 controls the invasive growth and virulence in *Fusarium oxysporum*. The Ste12 disruption mutants showed a substantial reduction in virulence when inoculated in common bean seedlings [21,22]. Similarly, *ΔPdSte12* mutants showed significantly reduced virulence during citrus fruit infection [23]. The Ste12 transcription factor of *Botrytis cinerea* was related to fungal germination and pathogenicity [24]. The essential role of the Ste12 transcription factor in the virulence of plant pathogenic fungi, such as *P. expansum* infecting apple fruit (Sánchez-Torres et al., 2018), the vascular pathogen *Verticillium dahliae*, and *Setosphaeria turcica* was already reported [25]. In the present study, the *Ste12* expression in *P. expansum* infecting pear fruit was up-regulated, which confirmed that *Ste12* might play an essential role in the infection progress [26].

The transcription factor MCM1 is a pheromone transcription factor in the MAPK signaling pathway. MCM1 was first found in *S. cerevisiae* and controlled numerous cellular and developmental processes of the yeast [27,28]. The FgMcm1 TF of *F. graminearum* plays a vital role in regulating cell identity, sexual and asexual reproduction, secondary metabolism and pathogenesis [29]. Likewise, the MoMcm1 TF of *Magnaporthe oryzae* is involved in male fertility, microconidium production, and virulence [30]. Similarly, *Sclerotinia sclerotiorum*’s TF SsMcm1 has also been linked to growth and virulence regulation [31].

Transcription factors SOK2, SIP4, ECM22, YLR278C and STP3 regulate the *WSC* family genes in the MAPK signaling pathway. In *S. cerevisiae*, a transcription factor cascade, Sok2 regulates cell-cell adhesion and yeast pseudohyphal differentiation [32]. The *WSC* family is a regulatory receptor upstream of the MAPK signaling pathway, which acts as mechanosensors of cell wall stress induced by wall remodeling during vegetative growth [33]. There are three *WSC* genes in *P. expansum*, two of which are involved in maintaining the integrity of the cell wall (ncbi_27683189 and ncbi_27672954). There is no relevant report on the functions of *WSC* genes in *P. expansum*, but it has been studied in other fungi. Gualtieri et al. (2004) reported that the cell wall sensor *Wsc1p* significantly regulates actin network rearrangements during membrane stretching and cell wall expansion [34]. The double and triple deletion mutants of *WSC1,2,3* showed severe cell integrity defects in *Kluyveromyces lactis* [35]. In *Aspergillus nidulans*, the putative stress sensors *WscA* and *WscB* were involved in cell wall integrity under hypo-osmotic and acidic pH conditions [36]. In *Neurospora crassa*, *WSC-1* played a significant part in cell wall integrity maintenance [37]. ATAC-seq analysis and RT-qPCR results of the present study confirmed the up-regulated expression of *WSC1,2,3* during pear fruit infection, which witnessed the crucial role of *WSC* genes in *P. expansum* infection progress.

In the pentose and glucuronate interconversions pathway, the expression of *PG1* in *P. expansum* was up-regulated. Polygalacturonase PG1 causes tissue leaching and protoplast death by degrading the homologous polygalacturonic acid region in plant cell walls and is closely related to the pathogenicity and toxicity of fungi [38,39,40]. Oeser et al. reported that polygalacturonase could be a pathogenicity factor in the *Claviceps purpurearye* interaction [41]. Likewise, a specific polygalacturonase, P2c, contributes to the invasion and spread of *Aspergillus flavus* during the infection of cotton bolls [42]. The *Bcpg1* gene of *B. cinerea* is required for its full virulence on tomato leaves and fruits and on apple fruits [39].

## 5. Conclusions

The transcription factors of *P. expansum* related to pear fruit infection were screened using ATAC-seq analysis. The differentially expressed peak-related genes in *P. expansum* were analyzed by GO enrichment analysis and KEGG pathway analysis. The specific roles of the transcription factors MCM1, Ste12, and *WSC* genes in the MAPK signaling pathway, *PG1* and *RPE1* in the pentose and glucuronate interconversions pathway, and their related genes play in the *P. expansum* infection of pear fruit were not witnessed, so far. Our results provide fundamental information about the critical transcription factors involved in *P. expansum* infection process; further research is warranted to explore the specific functions and characterization of these screened transcription factors.

## Figures and Tables

**Figure 1 foods-11-03855-f001:**
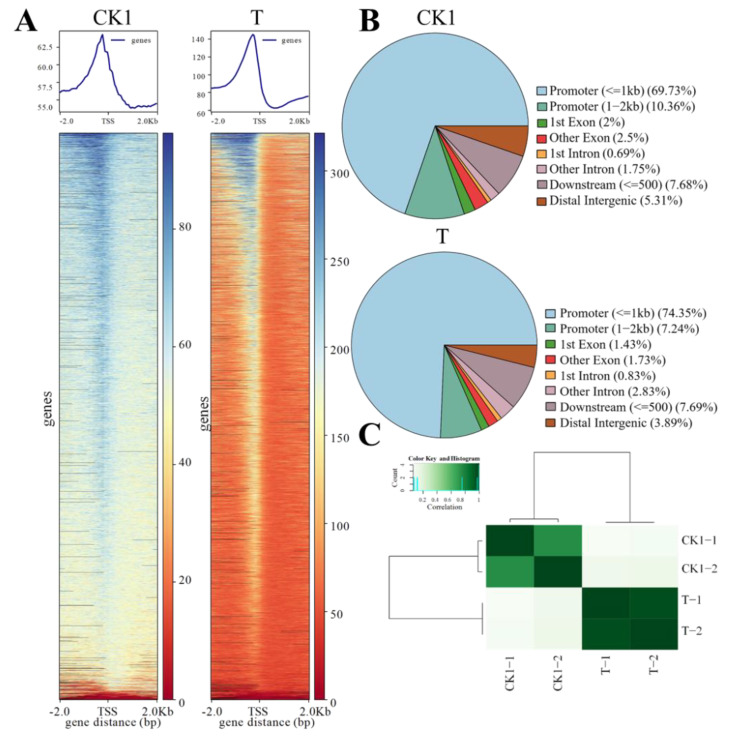
Chromatin accessibility landscape of CK1 and T samples. (**A**) Average plots and heat maps showing the signals at the TSS in the ATAC-seq data sets. (**B**) Distribution of peak in each functional region of the gene in CK1 and T samples. (**C**) Heatmap showing the Pearson correlation coefficient of two ATAC-seq replicates of two group samples.

**Figure 2 foods-11-03855-f002:**
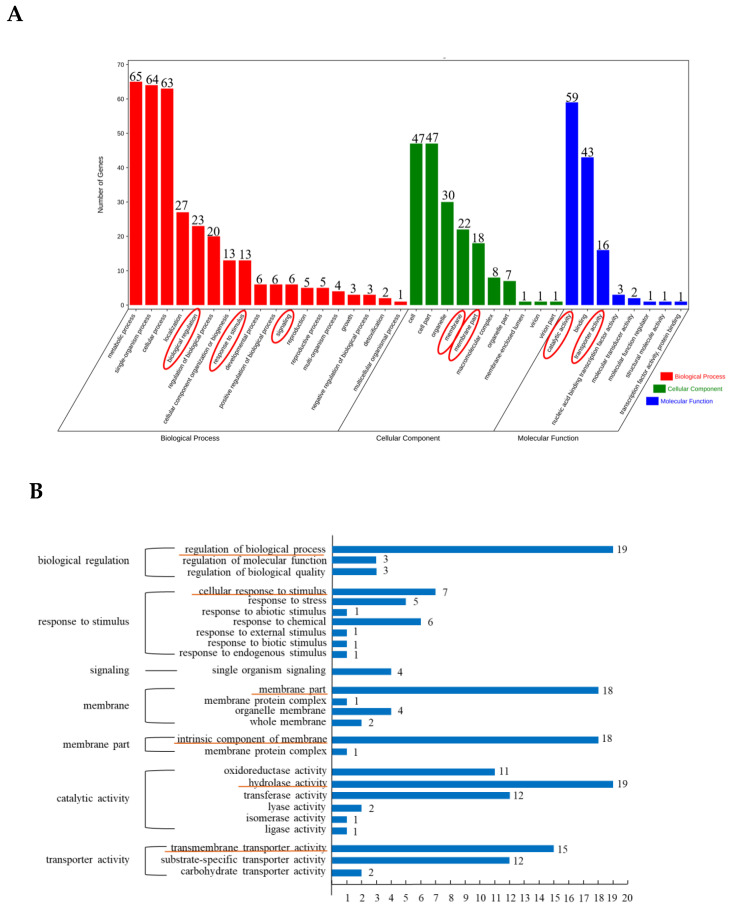
GO secondary (**A**) and tertiary (**B**) enrichment analysis of DEGs in *P. expansum.* The red circle in (**A**) and the yellow underline in (**B**) are GO secondary term and tertiary term related to infection.

**Figure 3 foods-11-03855-f003:**
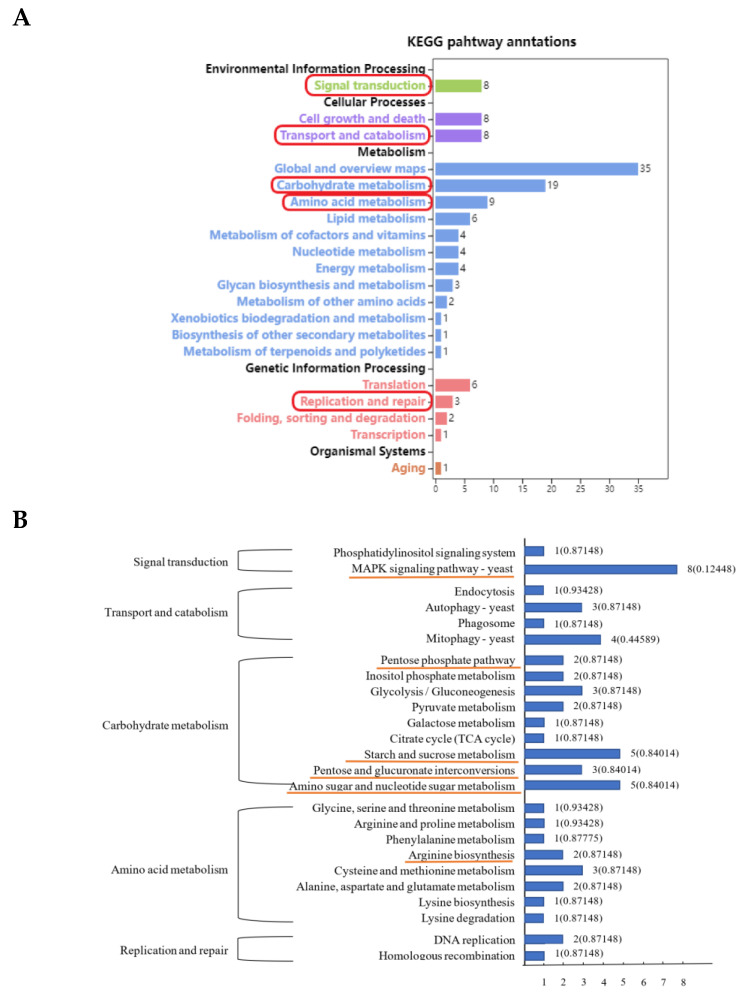
KEGG enrichment analysis of DEGs (**A**) and KEGG secondary subclass corresponding pathway of DEGs (**B**) in *P. expansum*. The red circle in (**A**) and the yellow underline in (**B**) marked are KEGG secondary term and pathways related to infection.

**Figure 4 foods-11-03855-f004:**
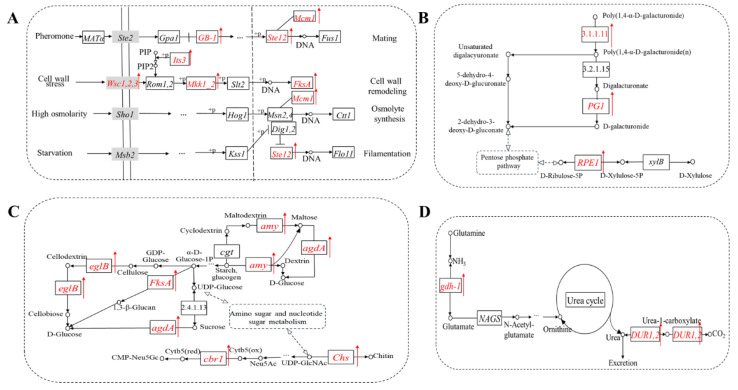
Several important KEGG pathways of DEGs in *P. expansum*. (**A**) MAPK signaling pathway; (**B**) Pentose and glucuronate interconversions & Pentose phosphate pathway; (**C**) Starch and sucrose metabolism & Amino sugar and nucleotide sugar metabolism; (**D**) Arginine biosynthesis. The red-colored gene names represent the up-regulated genes. Box represents gene (protein or mRNA), circle represents compound (generally metabolite).

**Figure 5 foods-11-03855-f005:**
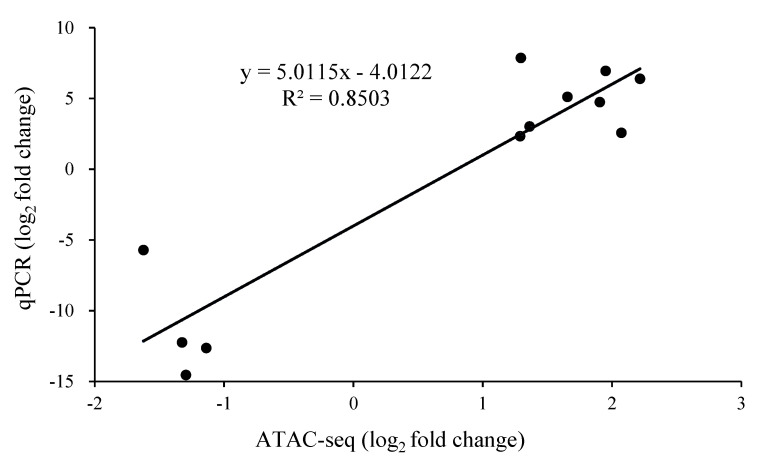
The comparison of differentially expressed peak-related genes in *P. expansum* between RT-qPCR and ATAC-seq data for the 12 selected genes. Fold-change values were converted to log2 (fold-change).

**Table 1 foods-11-03855-t001:** The primers used in RT-qPCR analysis.

Gene Name	Symbol	Primer	Sequence(5′-3′)	Length (mer)	Tm (°C)
ncbi_27683293	*sebA*	XM_016747872.1-F	CACAGCCTTGAAGACTCGGATGAC	24	60.2
		XM_016747872.1-R	TGCTACGCTTCTTGATCTTGGGTTC	25	59.3
ncbi_27675759	*Pma2*	XM_016740340.1-F	CTGGATTCTCGCTGCCTGGATTG	23	60.9
		XM_016740340.1-R	AGTGGAGCAGGTTGTTGTCGTTATC	25	59.2
ncbi_27674750	*isp4*	XM_016739331.1-F	CGGAGGAGATTGTCACGCATGTC	23	60.7
		XM_016739331.1-R	CACCGCAATGGAAGGGCTTACC	22	61.7
ncbi_27683189	*WSC*	XM_016747768.1-F	AACAGACCTCGAAGAGCAAGAACAC	25	59.1
		XM_016747768.1-R	CCAGAAGAAGATAGCACCGCAGAG	24	59.9
ncbi_27683152	*fksA*	XM_016747731.1-F	TGCCTTCTGGTTCTTCACTGCTTAC	25	59.2
		XM_016747731.1-R	CGGCGAGATATTTGGGCGAGTC	22	61.1
ncbi_27682174	*PG1*	XM_016746754.1-F	CGAGCAGGATTATCAGAACGGTAGC	25	59.6
		XM_016746754.1-R	CGCAGAGAATGTAGACGGGAATAGC	25	59.6
ncbi_27677161	*Ste12*	XM_016741742.1-F	CAGGAACTCGCTCGCACTTACC	22	60.9
		XM_016741742.1-R	CTGGATGTTGTACGGTGGTCTGATC	25	59.6
ncbi_27682273	*Mkk1_2*	XM_016746853.1-F	TCGCACCACATCTGTCGTTACTATG	25	58.7
		XM_016746853.1-R	GATGCTGTCCAGACTTCCACCTTC	24	60.1
ncbi_27683634	*adhB*	XM_016748213.1-F	GAATTTGGCAATTTGGGAGGCTGTC	25	59.3
		XM_016748213.1-R	TGTTGAGAATGGCGTGGCTAGTTAG	25	59
ncbi_27676242	*catA*	XM_016740823.1-F	CGTCAGCTACCCGCAGAAACAC	22	61.4
		XM_016740823.1-R	CGGCGAGAGCGAGTTGTAGAATAC	24	59.9
ncbi_27678339	*cetA*	XM_016742920.1-F	CATCAAGATGTCCACCAGCGAGAG	24	60.1
		XM_016742920.1-R	GGTGAAGCCCGACTTGACAAACTC	24	60.8
ncbi_27674565	*YFL054C*	XM_016739146.1-F	CATTGTTGAAGCCGCCACATTAGC	24	59.6
		XM_016739146.1-R	GAAGAACGCAGTCGCAGGATGG	22	61.5
ncbi_27673157	*DOA4*	XM_016737738.1-F	GGAAGACTCGCTGACGGAAGAAC	23	60.4
		XM_016737738.1-R	TTCAGGGAACCGACGCAAGAAAG	23	60.1

**Table 2 foods-11-03855-t002:** The information on the transcription factors regulating genes.

TF	TF Description	Gene ID	Symbol	Gene Description
HAP3	Heteromeric CCAAT-binding factors	ncbi_27674657	*its3*	1-phosphatidylinositol-4-phosphate 5-kinase
		ncbi_27677161	*Ste12*	transcription factor STE12
		ncbi_27679022	*MCM1*	pheromone receptor transcription factor
		ncbi_27682441	*agdA*	alpha-glucosidase
SOK2	APSES-type DNA-binding domain	ncbi_27674657	*its3*	1-phosphatidylinositol-4-phosphate 5-kinase
		ncbi_27682445	*DUR1,2*	urea carboxylase / allophanate hydrolase
		ncbi_27683189	*WSC*	cell wall integrity and stress response component
SIP4	C_6_ zinc cluster factors	ncbi_27679542	*RPE1*	ribulose-phosphate 3-epimerase
		ncbi_27683189	*WSC*	cell wall integrity and stress response component
YPR196W	C_6_ zinc cluster factors	ncbi_27678364	*eglB*	endoglucanase
		ncbi_27677161	*Ste12*	transcription factor STE12
ECM22	C_6_ zinc cluster factors	ncbi_27679542	*RPE1*	ribulose-phosphate 3-epimerase
		ncbi_27683189	*WSC*	cell wall integrity and stress response component
YLR278C	C_6_ zinc cluster factors	ncbi_27679542	*RPE1*	ribulose-phosphate 3-epimerase
		ncbi_27683189	*WSC*	cell wall integrity and stress response component
SUT2	C_6_ zinc cluster factors	ncbi_27679542	*RPE1*	ribulose-phosphate 3-epimerase
STP3	C_2_H_2_ zinc finger factors	ncbi_27672954	*WSC*	cell wall integrity and stress response component
YNR063W	C_6_ zinc cluster factors	ncbi_27677161	*Ste12*	transcription factor STE12

## Data Availability

Data is contained within the article.

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
