# Peer review of "Screening and Regulation Mechanism of Key Transcription Factors of Penicillium expansum Infecting Postharvest Pears by ATAC-Seq Analysis"

_foods, 2022, doi:10.3390/foods11233855_

Round 1

Reviewer 1 Report

In general, the manuscript is well written and shows interesting results and discussion.

Suggestions:

- Add the amount of pears lost annually (in ton) or the value of economic loss ($) due to blue mold. If this value has not yet been estimated for blue mold, add information on the pears loss in relation to postharvest diseases, please. 

- Add full name of some abbreviations or acronyms in the fist time mentioned in the manuscript. Examples:

ATAC-Seq - assay for transposase-accessible chromatin with sequencing (ATAC-Seq) (line 79, Introduction);

GO enrichment - Gene Ontology (GO) enrichment (line 79, Introduction); and so on... 

- Increase the font size in legends of Figure 1, Figure 2A, and Figure 4, please. 

Minor Corrections:

"...with the infection of pear fruit by P. expansum.." (line 29, Abstract) - Delete an endpoint.

Author Response

Reviewer #1:

  1. Add the amount of pears lost annually (in ton) or the value of economic loss ($) due to blue mold. If this value has not yet been estimated for blue mold, add information on the pears loss in relation to postharvest diseases, please.

Response:

Thanks for your kind advices, we have added it on Line 40.

  1. Add full name of some abbreviations or acronyms in the first time mentioned in the manuscript.

Response:

Thanks for your kind advice, we have added full name of some abbreviations or acronyms in the first time mentioned in the manuscript.

  1. Increase the font size in legends of Figure 1, Figure 2A, and Figure 4, please.

Response:

Thanks for your question, the font size in legends of Figure 1 and Figure 4 have been increased, and the original font size can only be maintained due to typesetting in Fig. 2A.

  1. "...with the infection of pear fruit by P. expansum.." (line 29, Abstract) - Delete an endpoint.

Response:

Thanks for your kind reminder, the endpoint has been deleted.

Reviewer 2 Report

Study sounds well, executed in planned way, results has significance for further research in the area, however English is bit complicated and strange some time , needs extensive improvement.

Improve the paper in light of suggestion given on the body of the MS

Author Response

Reviewer #2: Study sounds well, executed in planned way, results has significance for further research in the area, however English is bit complicated and strange some time , needs extensive improvement.

Improve the paper in light of suggestion given on the body of the MS.

  1. Line 16-20, May not be the part of Abstract.

Response:

Thanks for your kind advice, we have appropriately deleted the background introduction.

  1. Line 20, "this study" should be capitalized and line 29, delete an endpoint.

Response:

Thanks for your kind reminder, "t" has been capitalized and the endpoint has been deleted.

  1. Some words need to be replaced. For example, Line 30, "verify"; line 32, "12"; line 38, "loved by people"; line 50, "infecting" and so on.

Response:

Thanks for your suggestion, we have revised and marked them in red color as you pointed out.

  1. Some words need to be deleted, for example, Line 32, "results"; line 37, "and"; line 40, "in the pear fruit industry" and so on.

Response:

Thanks for your suggestion, we have revised them as you suggested.

  1. Line 43 and 47, relevant literature needs to be added.

Response:

Thanks for your suggestion, we have revised them on Line 45-47.

  1. Line 55, is it a technical term to describe mentioned protein?

Response:

Thanks for your question, it is a technical term to describe mentioned protein.

  1. Some sentences and paragraphs need to be revised, like line 61-64, line 70, line 311-315, line 323-325, line 331-335 and conclusion.

Response:

Thanks for your suggestion, based on your suggestions, we have revised the manuscript and marked them in red.

  1. Line 113 and 132, Sub title number needs to be added.

Response:

Thanks for your kind reminder, sub title number have been added as you pointed.

  1. Line 268, assure readability of these graphs.

Response:

Thanks for your suggestion, the font size in legends of Figures have been increased.

Also the manuscript was revised by a foreign teacher (Solairaj Dhanasekaran) of our research team, whose mother language is English, we have marked them in red in the manuscript.